# Downfolding the Su-Schrieffer-Heeger model

Arne Schobert[1], Jan Berges[1], Tim Wehling[1] and Erik van Loon[1,2*]

**1** Institut für Theoretische Physik, Bremen Center for Computational Materials Science, and MAPEX Center for Materials and Processes, Universität Bremen, Bremen, Germany
**2** Department of Physics, Lund University, Lund, Sweden

⋆ erik.van_loon@teorfys.lu.se

## Abstract

Charge-density waves are responsible for symmetry-breaking displacements of atoms and concomitant changes in the electronic structure. Linear response theories, in particular density-functional perturbation theory, provide a way to study the effect of displacements on both the total energy and the electronic structure based on a single *ab initio* calculation. In downfolding approaches, the electronic system is reduced to a smaller number of bands, allowing for the incorporation of additional correlation and environmental effects on these bands. However, the physical contents of this downfolded model and its potential limitations are not always obvious. Here, we study the potential-energy landscape and electronic structure of the Su-Schrieffer-Heeger (SSH) model, where all relevant quantities can be evaluated analytically. We compare the exact results at arbitrary displacement with diagrammatic perturbation theory both in the full model and in a downfolded effective single-band model, which gives an instructive insight into the properties of downfolding. An exact reconstruction of the potential-energy landscape is possible in a downfolded model, which requires a dynamical electron-biphonon interaction. The dispersion of the bands upon atomic displacement is also found correctly, where the downfolded model by construction only captures spectral weight in the target space. In the SSH model, the electron-phonon coupling mechanism involves exclusively hybridization between the low- and high-energy bands and this limits the computational efficiency gain of downfolded models.



# 1 Introduction

The study of electron-phonon interactions (EPIs) goes back to the early days of solid-state theory. They are important for our understanding of basic material properties such as effective masses [1–4] and lattice constants [5,6]. Furthermore, this interaction is responsible for phase transitions, such as conventional superconductivity [7–18] and charge-density waves (CDWs) [19–30]. Even in unconventional superconductors, signatures of EPIs can be found [31–43]. However, the precise interplay responsible for these phenomena is not fully understood, which is one of the reasons that the fundamental interaction between electrons and phonons needs to be described accurately. Developments in this direction occur along two main paths: first-principles calculations and model Hamiltonians.

The standard *ab initio* method for calculating the EPI is the density-functional perturbation theory (DFPT) [44]. The most important ingredients of this theory are the adiabatic Born-Oppenheimer approximation [45], density-functional theory (DFT) [46], and linear-response theory. Briefly put, these state that it is possible to separate the dynamics of the electrons and ions, treat the electron in an effective one-body Schrödinger equation, and calculate the response of the electrons upon displacement of the nuclei within linear order, based only on the electronic density [47–49]. The resulting EPI simultaneously describes two sides of the same coin, namely how the electrons screen and renormalize the phonons and how the electronic structure will adjust to atomic displacements. For an overview of the historical development and the recent accomplishments of calculating the EPI from first principles, see Ref. [50].

Despite the unquestionable success of the current *ab initio* computational methods, another trend in the literature is to treat the important physical phenomena in correlated materials with *downfolding* approaches. The central idea is to reduce the number of degrees of freedom compared to the full system by keeping only the relevant states in a low-energy theory. The other states are integrated out and determine the parameters of the downfolded system. The overarching purpose of this procedure is the application of more advanced and expensive computational techniques only to the low-energy space where correlations take place.

For phonon-related properties, the *constrained* density-functional perturbation theory

(cDFPT) was introduced [51] and successfully applied to superconducting materials such as alkali-doped fullerides [52] and light elements [53]. Additionally it was applied to monolayer 1H-TaS$_2$ [29], where it was shown that the CDW in this material is induced by coupling between the longitudinal-acoustic phonons and the electrons from an isolated low-energy metallic band. With the help of cDFPT it is possible to extract unscreened or partially screened parameters such as the phonon frequency and the electron-phonon vertex from an *ab initio* calculation and use these as the basis for an effective low-energy model Hamiltonian. The usefulness of partially screened parameters lies in the fact that they get rid of the coupling between phonons and the high-energy electrons.

As discussed, the description of real physical phenomena that are tightly linked to the EPIs is frequently based on *ab initio* theories (DFPT, cDFPT) that involve substantial numerical and computational effort. The structure of the theory is not always transparent, and also obscured by details of the numerical implementation. To avoid these complications, a second branch in the literature is focused on model Hamiltonians. The most popular models of the EPI are the Fröhlich model [54] for polaron formation, the Holstein model [55] for optical phonons, and the Su-Schrieffer-Heeger (SSH) model [56] for CDWs.

For understanding the interplay of electronic structure and atomic displacements, the SSH model is the most instructive since it explicitly describes how the electronic band structure is renormalized by the displacements of the atoms. Previous investigations using this model have studied properties such as the effective mass [57, 58] and the band structure [59, 60], but also phonon-related properties [61]. In the model, a periodic displacement of the atoms can open a band gap and thereby lower the total energy of the system [56], leading to a CDW transition. In other words, electronic screening makes the CDW phonon go soft. This textbook example of a CDW transition [62] is appealing for the investigation of downfolding since it is possible to perform all calculations exactly once the Born-Oppenheimer approximation has been applied.

The origin of this extraordinary simplicity lies in the observation that the Born-Oppenheimer approximation makes the phonons classical and the remaining electronic degrees of freedom in the SSH model are noninteracting. Thus, given any fixed displacement, the resulting electronic Hamiltonian is easily diagonalized. In some sense, this is similar to the method employed in Hirsch-Fye Quantum Monte Carlo [63], where a Hubbard-Stratonovich transformation is used to generate a system of noninteracting electrons coupled to classical fields and the subsequent analysis only involves varying the classical field and evaluating the noninteracting electron system. Unlike in Hirsch-Fye Quantum Monte Carlo, here the classical field is directly observable and has a clear physical meaning.

We choose to study the SSH model here for its simplicity, acting as a minimal model for electron-phonon coupling. At the same time, this means that there are many relevant aspects of electron-phonon coupling and CDWs that are not captured by the SSH model. In particular, the SSH model neglects Coulomb interactions between the electrons, and these are responsible for important effects such as screening and entirely electronic CDWs without lattice displacement. Furthermore, in higher dimensions, the shape of the Fermi surface can play an important role, in the form of nesting and Van Hove singularities. Given the complexity of electron-phonon systems, studying simple models is a useful way to identify relevant effects and mechanisms.

In this work, we compare the direct calculation of properties of the SSH model in the Born-Oppenheimer approximation at finite displacement with a perturbative diagrammatic expansion around the undistorted state à la DFPT. In this model, the diagrammatic expansion can be evaluated analytically order by order and we show that it correctly captures how the electron-phonon coupling renormalizes the phonon frequency and the electronic structure. Then, in the spirit of downfolding, we move to an effective single-band model for the dimerization transition in the SSH model. The diagrammatic structure in this effective model differs

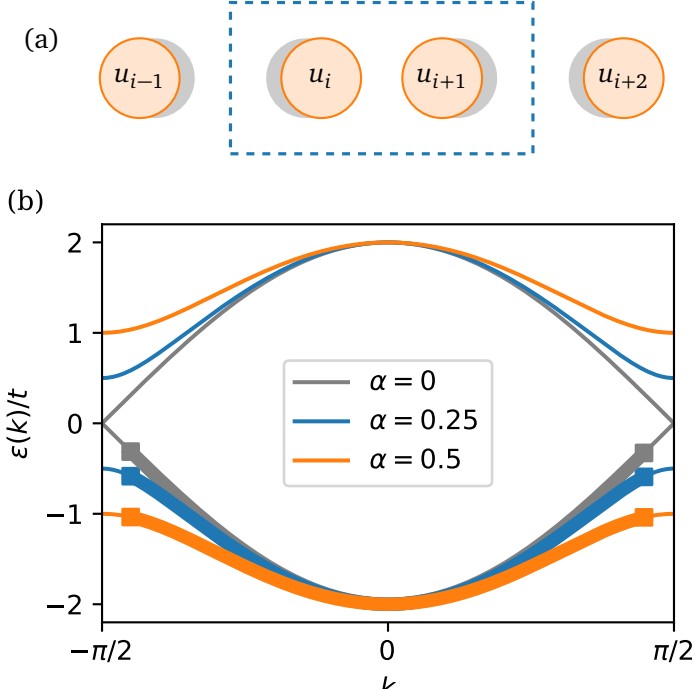

Figure 1: (a) Dimerization. (b) Band structure at various values of the atomic displacement $\alpha$. The thick lines represent the occupied states when there are $\langle n \rangle = 0.9$ spinless electrons per dimer.

substantially from the original model: an interaction between an electron and two phonons appears and this interaction turns out to be dynamical with a frequency set by the high-energy electrons that were integrated out. We show that this downfolded model faithfully reproduces the energy landscape and the CDW. Furthermore, we discuss a cDFPT-like approach to down-folding, which correctly describes the screening of the phonon frequencies. In the SSH model, the displacement-induced orbital reconstruction between target and rest space is the central aspect of the downfolding and there is no remaining electron-phonon coupling in the cDFPT low-energy model.

## 2 Model

In this work, we consider the SSH model [56] in the classical Born-Oppenheimer limit [64], i.e., we ignore the kinetic energy of the atoms. We consider spinless fermions in a one-dimensional lattice with Hamiltonian

$$H = -t \sum_{i=0}^{N-1} (1 + u_i - u_{i+1})(c_i^\dagger c_{i+1} + c_{i+1}^\dagger c_i) + \frac{k_s}{2} \sum_{i=0}^{N-1} (u_{i+1} - u_i)^2. \tag{1}$$

Here, $u_i$ is a (classical) variable describing the atomic displacements, with $0 \leq i < N$. We use the periodic boundary condition $u_N \equiv u_0$. The hopping $t > 0$ sets the electronic energy scale and the force constant $k_s > 0$ that of the phonons.

We consider dimerization, i.e., displacements of the form $u_i = (-1)^i \alpha/2$, and double the unit cell to include entire dimers. This is illustrated in Fig. 1a. Using the notation $a_i = c_{2i}$ and

$b_i = c_{2i+1}$, we obtain

$$H = -t \sum_{i=0}^{N/2-1} (1+\alpha)(a_i^\dagger b_i + b_i^\dagger a_i) - t \sum_{i=0}^{N/2-1} (1-\alpha)(a_{i+1}^\dagger b_i + b_i^\dagger a_{i+1}) + \frac{1}{2} N k_s \alpha^2. \tag{2}$$

Performing a Fourier transform to momentum space, the Hamiltonian in matrix form reads

$$H = \sum_k \begin{pmatrix} a_k^\dagger & b_k^\dagger \end{pmatrix} \hat{\varepsilon}(k) \begin{pmatrix} a_k \\ b_k \end{pmatrix} + \frac{1}{2} N k_s \alpha^2, \tag{3}$$

$$\hat{\varepsilon}(k) = -t \begin{pmatrix} 0 & 1+\alpha+(1-\alpha)e^{2ik} \\ 1+\alpha+(1-\alpha)e^{-2ik} & 0 \end{pmatrix}, \tag{4}$$

with eigenvalues

$$\varepsilon_\pm(k) = \pm 2t \sqrt{1 + (\alpha^2-1)\sin^2(k)} = \pm 2t \sqrt{\cos^2(k) + \alpha^2 \sin^2(k)}. \tag{5}$$

These give the dispersion shown in Fig. 1b. Note that the Brillouin zone is $-\pi/2 \le k \le \pi/2$, where $k$ is made dimensionless by setting the atomic distance to unity.

In the following, we assume that the electronic density $\langle n \rangle$ is smaller than 1 electron/dimer. Since the model is particle-hole symmetric, the case $\langle n \rangle > 1$ follows by symmetry. The situation $\langle n \rangle = 1$ (half-filling) is special and will be discussed in more detail below, see Sec. 7. At zero temperature, the electron density is proportional to the Fermi wave vector $k_f$ and independent of $\alpha$: $\langle n \rangle = 2k_f/\pi$. The total electronic energy per dimer, in the thermodynamic limit $N \to \infty$, is

$$E_{\text{el}} = \frac{1}{\pi} \int_{-k_f}^{k_f} \varepsilon_-(k) dk, \tag{6}$$

and the total energy per dimer is

$$E = k_s \alpha^2 + E_{\text{el}}. \tag{7}$$

Note that in this model, displacements do not change the Fermi surface and the electronic energy $E_{\text{el}}$ depends on $\alpha$ only via Eq. (5), which will allow us to pull derivatives through the integral in Eq. (6).

In Fig. 2a, we show how the total energy depends on $\alpha$ for fixed $k_s$ and $\langle n \rangle$. The total energy is obviously symmetric in $\alpha$, and the undistorted lattice at $\alpha = 0$ is an extremum of the total energy. Without electrons, $E_{\text{bare}} = k_s \alpha^2$ is a convex parabola with a minimum at $\alpha = 0$. However, the coupling to the electrons can lead to a Peierls CDW phase transition where $\alpha = 0$ turns into a local maximum and two global minima occur at $\alpha = \pm \alpha^*$. The finite $\alpha$ lowers the energy of the occupied states and thus the total electronic energy and this compensates for the gain in potential energy due to $\alpha$.

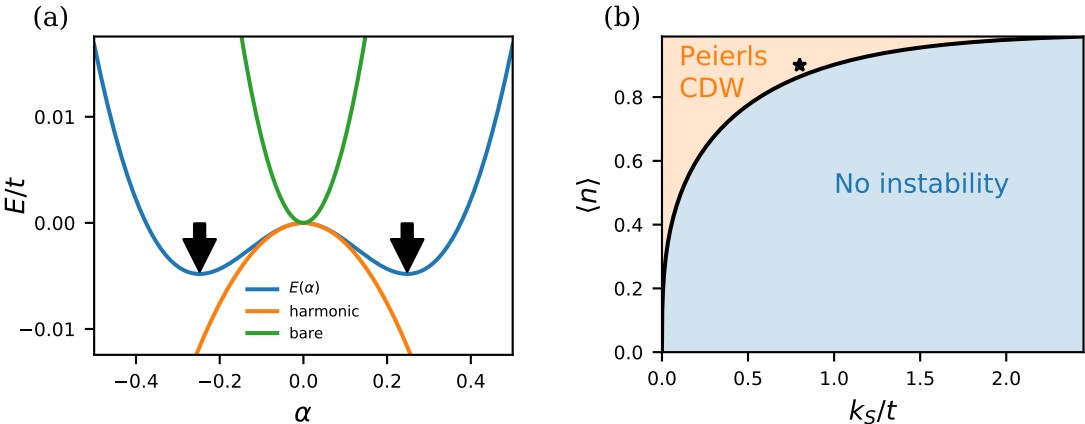

Figure 2: (a) Energy landscape at $k_s/t = 0.8$, $\langle n \rangle = 0.9$. The curves show the exact energy $E(\alpha)$, the harmonic approximation including electronic screening $E(0) + \frac{1}{2}\omega^2\alpha^2$, and the bare phonon energy $E(\alpha = 0) + \frac{1}{2}\omega_{\text{bare}}^2\alpha^2$. The arrows indicate the minima at $\pm\alpha^*$. (b) Phase diagram of the SSH model for the density $\langle n \rangle$ and the force constant $k_s$. The black star marks the parameters of (a). We only consider the transition to the dimerized CDW.

## 3 Harmonic and anharmonic lattice potential

To analyze the phase transition, it is useful to perform a Taylor expansion of the lattice potential $E(\alpha)$ around $\alpha = 0$.

$$E(\alpha) - E(0) = \frac{1}{2} \left.\frac{d^2 E(\alpha)}{d\alpha^2}\right|_{\alpha=0} \alpha^2 + \frac{1}{4!} \left.\frac{d^4 E(\alpha)}{d\alpha^4}\right|_{\alpha=0} \alpha^4 + \dots \tag{8}$$

$$\equiv \frac{1}{2}\omega^2\alpha^2 + h^{(4)}\alpha^4 + \dots \tag{9}$$

$$\omega^2 = \omega_{\text{bare}}^2 + \Delta\omega^2, \tag{10}$$

$$\omega_{\text{bare}}^2 \equiv 2k_s, \tag{11}$$

$$\Delta\omega^2 \equiv \frac{1}{\pi} \int_{-k_f}^{k_f} dk \left.\frac{d^2\varepsilon_-(k)}{d\alpha^2}\right|_{\alpha=0} = -\frac{2t}{\pi} \int_{-k_f}^{k_f} dk \frac{\sin^2(k)}{\cos(k)}, \tag{12}$$

$$h^{(4)} = \frac{1}{\pi} \int_{-k_f}^{k_f} dk \frac{1}{4!} \left.\frac{d^4\varepsilon_-(k)}{d\alpha^4}\right|_{\alpha=0} = \frac{t}{4\pi} \int_{-k_f}^{k_f} dk \frac{\sin^4(k)}{\cos^3(k)}. \tag{13}$$

Here, we have introduced the bare phonon frequency $\omega_{\text{bare}}$ and the dressed phonon frequency $\omega$. The difference $\Delta\omega^2$, the electronic screening of the phonon, originates in the change in electronic structure in response to the lattice distortion. Screening lowers the phonon frequency, and the Peierls transition occurs when the dressed phonon frequency is equal to zero, i.e., $\omega = 0$. In Fig. 2b, the Peierls transition is represented as the black line that separates the phases $\omega^2 < 0$ (Peierls instability) and $\omega^2 > 0$ (no instability). As we can see, a weak force constant $k_s$ and a density $\langle n \rangle$ close to half-filling is preferred for a Peierls instability. Beyond the Peierls transition, $\alpha = 0$ is a local maximum of the potential and the higher-order terms, such as $h^{(4)}$, are responsible for ensuring that $E(\alpha)$ has a minimum at some finite $\alpha$. In Appendix A, we show that there can be at most two minima, symmetrically located around $\alpha = 0$. Only even orders of $\alpha$ appear due to the symmetry of the system.

# 4 Electron-phonon coupling: Two-band model

In the previous section, we used our knowledge of the exact dependence of the electronic structure $\hat{\varepsilon}$ on $\alpha$ to determine the potential-energy landscape. In *ab initio* calculations (e.g., DFPT), one will usually not have access to this. Instead, the only known quantities are the electronic structure of the undistorted structure $\hat{\varepsilon}_0$ and the electron-phonon coupling, the first derivative of the electronic structure with respect to the displacement. Access to the latter quantity is guaranteed by the $2n + 1$ theorem [47–49]. Because of this, it is instructive to calculate the (approximate) potential-energy landscape of the SSH model—and in particular the screening of the phonon frequency—based just on these quantities in a perturbative expansion around $\alpha = 0$.

The Feynman rules can be read off from the Hamiltonian, Eq. (3), by writing it as

$$\hat{H} = \sum_{k} f_k^{\dagger} \hat{\varepsilon}_0(k) f_k + \alpha f_k^{\dagger} \underline{\hat{g}}(k) f_k + N \frac{1}{2} \omega_{\text{bare}}^2 \alpha^2. \tag{14}$$

Here, $f^{\dagger}$ is shorthand for the vector $(a^{\dagger}, b^{\dagger})$. There is a single $q = 0$ phonon mode corresponding to dimerization, with frequency $\omega_{\text{bare}}^2 = 2k_s$. This mode is entirely classical, since we are interested only in a Born-Oppenheimer potential-energy landscape. The electron-phonon coupling is a matrix in electronic space and is obtained as $\underline{\hat{g}} = d\hat{\varepsilon}/d\alpha$ evaluated at $\alpha = 0$. In other words, it consists of the parts of $\hat{\varepsilon}$ that are proportional to $\alpha$. Explicitly,

$$\underline{\hat{g}}(k) = -t \begin{pmatrix} 0 & 1 - e^{2ik} \\ 1 - e^{-2ik} & 0 \end{pmatrix} \text{ in the } (a^{\dagger}, b^{\dagger}) \text{ basis.} \tag{15}$$

Note that we are considering a single phonon mode at $q = 0$, so we do not need a $q$ label on $\underline{\hat{g}}$. The lack of higher-order electron-phonon-coupling terms in Eq. (14) is a special property of the SSH model.

To evaluate the Feynman diagrams, it is most convenient to express the electronic part of the Hamiltonian in the eigenbasis of the unperturbed electronic system. This basis transformation can be seen in Appendix B. The transformed electron-phonon coupling is

$$\hat{g}(k) = 2t \begin{pmatrix} 0 & i \sin(k) \\ -i \sin(k) & 0 \end{pmatrix} \text{ in the band basis.} \tag{16}$$

We observe that $g$ couples the two bands and has no intraband component. In other words, to linear order in $\alpha$ around $\alpha = 0$, distortions only change the orbital composition of the bands but not the dispersion of the bands.

The vanishing diagonal elements of $g$ can also be understood as a symmetry selection rule. The inversion symmetry of the system implies that $\varepsilon(\alpha)$ and $\varepsilon(-\alpha)$ have the same eigenvalues and this implies both $\text{Tr}\, g = \text{Tr}\, \frac{d\hat{\varepsilon}}{d\alpha} = \frac{d}{d\alpha} \text{Tr}\, \hat{\varepsilon} = 0$, which holds in any basis, and $\langle n| \hat{g} |n\rangle = 0$ for any $\alpha = 0$ eigenvector $|n\rangle$, since these $|n\rangle$ are eigenvectors of the inversion operator with eigenvalue $\pm 1$.

## 4.1 Leading diagram

We are interested in establishing the effective potential felt by the atoms, including electronic screening. Diagrammatically, this means that the phonon mode only appears as external lines, whereas internally the diagram consists of electronic propagators and electron-phonon vertices. All diagrams with $n$ external lines need to be summed to obtain the $\alpha^n$ coefficient in the potential $E(\alpha)$.[1]

---

[1]For the diagrammatic expansion of the free energy, see Ref. [65].

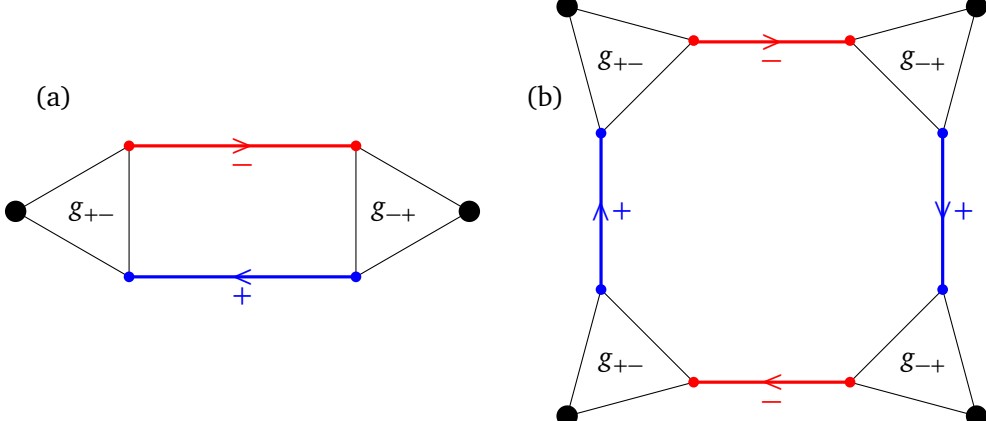

Figure 3: (a) Diagram for the renormalization of the phonon frequency. The black dots represent external phonon lines, the red and blue lines denote the electronic Green's functions $G_\pm$ in the band basis, and the triangles are the electron-phonon coupling. (b) Fourth order diagram.

For all upcoming diagrams, we will use the electronic Green's function

$$\hat{G}(E,k) = \frac{\hat{1}}{E\,\hat{1} - \hat{\varepsilon}_0(k) + i\hat{\eta}_k}, \tag{17}$$

where $\hat{1}$ is the identity matrix, the division denotes matrix inversion, and $\hat{\eta}_k$ denotes the usual small imaginary constant that is positive (negative) for empty (occupied) states, respectively.

For the phonon self-energy, i.e., with two external lines, there is only a single diagram, shown in Fig. 3a for one possible choice of the band indices, which corresponds to

$$\Delta\omega^2 = \sum_{m,n\in\{+,-\}} \int \frac{dk}{\pi}\, g_{m,n}(k)\Pi_{m,n}(k)g_{n,m}(k), \tag{18}$$

$$\Pi_{m,n}(k) = \frac{f_m(k) - f_n(k)}{\varepsilon_m(k) - \varepsilon_n(k)}, \tag{19}$$

$$f_m(k) = \begin{cases} 1 & \text{for } m = -1 \text{ and } |k| \le k_f, \\ 0 & \text{otherwise.} \end{cases} \tag{20}$$

This allows us to simplify the result to

$$\Delta\omega^2 = -\frac{2}{\pi}\int_{-k_f}^{k_f} dk\, \frac{|g_{+-}(k)|^2}{\varepsilon_+ - \varepsilon_-} = -\frac{2}{\pi}\int_{-k_f}^{k_f} dk\, \frac{4t^2\sin^2(k)}{4t\cos(k)} = -\frac{2t}{\pi}\int_{-k_f}^{k_f} \frac{\sin^2(k)}{\cos(k)}dk. \tag{21}$$

This is consistent with Eq. (12). This shows that the harmonic energy landscape can be calculated entirely from the undistorted structure at $\alpha = 0$, based on the electronic dispersion $\hat{\varepsilon}_0$ and the electron-phonon coupling $\hat{g}$.

## 4.2 Higher-order diagrams

It is also possible to calculate the energy landscape beyond the quadratic term. A special property of the SSH model is that the there are no higher-order electron-phonon vertices nor anharmonic bare phonon terms. Because of this, the entire perturbation theory is expressed in $\varepsilon_\pm$ and $g$. For example, the diagram for the fourth order contribution $\alpha^4$ is shown in Fig. 3b.

This is the only connected diagram at this order.[2] Note that all external phonons have $q = 0$, so all electronic lines have the same momentum $k$ and energy $E$. The band index of the electronic lines is alternating, since the electron-phonon coupling is entirely off-diagonal. The expression corresponding to this diagram is of the form

$$h^{(4)} = \frac{1}{2} \int \frac{dk}{\pi} \int dE \, g_{+-}(k)g_{-+}(k)g_{+-}(k)g_{-+}(k) \, G_-(k,E)G_+(k,E)G_-(k,E)G_+(k,E), \quad (22)$$

which already includes a factor 2 accounting for the fact that there is a second way to assign the band indices.[3]

The product of Green's functions can be reduced by repeated application of the relation $AB = (B - A)/(A^{-1} - B^{-1})$ for $A \neq B$, which is helpful because $G_\pm^{-1}(k,E) = E \mp |\varepsilon_0(k)| + i\eta_k$ is very simple. Below, all $G$'s have the same arguments $k, E$, which were dropped for notational convenience.

$$G_-G_+G_-G_+ = (G_+ - G_-)\frac{1}{2|\varepsilon_0|}(G_+ - G_-)\frac{1}{2|\varepsilon_0|} = \frac{G_-^2 + G_+^2}{4|\varepsilon_0|^2} - \frac{G_+ - G_-}{4|\varepsilon_0|^3}. \quad (23)$$

In the denominators we have already safely taken the limit $\eta \to 0$. Now, the integral over $E$ can be performed using $\int dE G_\pm^2(E) = 0$ and $\int dE G_\pm(E) = n(\varepsilon_\pm(k))$. Here, $n(\varepsilon_\pm(k))$ is the occupation, which is unity for the $-$ branch and $|k| < k_f$ and zero otherwise. This gives the same result as Eq. (13),

$$h^{(4)} = \frac{1}{2} \int_{-k_f}^{k_f} \frac{dk}{\pi} (2t)^4 \sin^4(k) \frac{1}{4(2t)^3 \cos^3(k)} = \frac{t}{4\pi} \int_{-k_f}^{k_f} dk \frac{\sin^4(k)}{\cos^3(k)}. \quad (24)$$

Diagrams at higher order can be evaluated in the same way, by repeated simplification of products of Green's functions. An interesting aspect is that the entire potential-energy landscape $E(\alpha)$ can be calculated in this way (for $\langle n \rangle \neq 1$, see Sec. 7) without ever determining how the band dispersion changes.

### 4.3 Change in electronic structure

The change in the electronic structure is given by the self-energy $\Sigma(E, k)$ and can be obtained diagrammatically by considering the sum of all one-electron irreducible diagrams. Now, the electronic lines are amputated and the phononic ends of the vertices are connected to crosses representing $\alpha$. This is similar to the way an external Zeeman magnetic field or scattering potential can be included in a diagrammatic theory. Note that due to the Born-Oppenheimer approximation, there is no true phonon propagator with two end points, which would represent the phonon dynamics.

In the present model, it turns out that there is only a single, trivial diagram for the self-energy,

$$\Sigma_{+-} = g_{+-}\alpha = \quad \text{[diagram]} \, \alpha \,, \quad (25)$$

with an equivalent diagram for $\Sigma_{-+}$. Together, they recover the exact electronic Green's function $\hat{\mathcal{G}}$ via the Dyson equation,

$$\hat{\mathcal{G}}^{-1} = \hat{G}^{-1} - \hat{\Sigma} = E - \hat{\varepsilon}_0 - \alpha\hat{g} + i\eta_k = E - \hat{\varepsilon} + i\eta_k. \quad (26)$$

---

[2]We remind the reader that we consider classical displacements, in the sense of the Born-Oppenheimer approximation. Thus, internal phonon propagators are not allowed in the diagrams.

[3]The $-$ line starting at the top left could also go to the bottom left instead of the top right. To keep the diagram connected, all other lines are then immediately fixed.

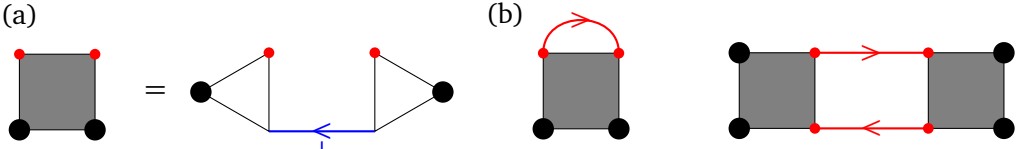

Figure 4: (a) The interaction vertex of the effective single-band theory (left-hand side) can be expressed in terms of the original vertices and the electronic band that is integrated out. (b) The diagrams responsible for the $\alpha^2$ and $\alpha^4$ contributions to the energy in the single-band model.

## 5 Single-band effective model

At $\langle n \rangle < 1$, there is only one partially filled band and this motivates us to investigate the possibility of describing the CDW via a single-band effective model. Here, we construct a model consisting of the partially filled electronic band, the bare phonon, and the coupling between the two. Formally, such a model is obtained by integrating out the unoccupied band of the two-band model. The effective action of the single-band model contains (partially) renormalized, dynamically screened interactions between these electrons and the phonons. In fact, the interaction vertices in this effective model can and do have an entirely different structure compared to those of the original two-band model. Generally, the vertices in the effective theory are obtained by collecting all connected diagrams consisting of rest space (here: $\varepsilon_+$) internal lines with a particular number of external phonon and target space (here: $\varepsilon_-$) lines, and an infinite set of vertices can appear in this way. The only general constraints are the conservation of the fermion number and momentum conservation. Thus, the low-energy Hamiltonian can contain interactions of the form $\alpha^m (c^\dagger c)^n$ for arbitrary $m$ and $n$. However, additional symmetries of the system can provide further constraints on the effective action.

Here, the single-band model is energetically completely symmetric in $\alpha \leftrightarrow -\alpha$ and this implies that only even powers of $\alpha$ can appear in the effective action. In other words, only interaction vertices with an even number of phonon lines are allowed.[4]

In fact, looking at the diagrammatic structure, it turns out that the single-band effective theory of the SSH model only contains one interaction vertex, shown in Fig 4a. This vertex has two phonon and two external electronic lines (one incoming, one outgoing) and takes the value

$$V(E,k) = |g_{+-}|^2 \, G(E,k) = 4t^2 \sin^2(k) \frac{1}{E - |\varepsilon_0(k)| + i\eta_k}. \tag{27}$$

Note that $V$ depends explicitly on $E$; the screened interactions that enter the effective model are dynamical quantities. The effective model contains only a single fermion with dispersion $\varepsilon_-$, so no further electronic band label is necessary.

The downfolded SSH model has only a single effective interaction vertex. This happens because the electron-phonon coupling in the original SSH model only has a single external high-energy electron (blue line in Fig. 4a). On the other hand, if the original model had contained either electron-electron interactions in the high-energy band or electron-phonon coupling between different electronic states in the high-energy band, then the downfolding would be more involved, since more diagrammatic contributions would appear in the expression for the effective action.

---

[4]Note that in the two-band model, although the eigenvalues are symmetric in $\alpha$, the eigenvectors are not and this leads to the finite value of $\hat{g}$, which is entirely off-diagonal in the electronic eigenbasis.

For the energy $E(\alpha)$, the second-order contribution, shown in Fig. 4b, is

$$\frac{1}{2}\Delta\omega^2 = \int \frac{dk}{\pi} \int dE\, V(E,k)G(E,k), \tag{28}$$

which upon insertion of Eq. (27) is equal to the result we obtained in the two-band model. Similarly, the fourth-order contribution, also shown in Fig. 4b, is

$$
\begin{aligned}
h^{(4)} &= \frac{1}{2} \int \frac{dk}{\pi} \int dE\, V^2(E,k)G^2(E,k) \\
&= \frac{1}{2} \int \frac{dk}{\pi} \int dE \frac{16t^4 \sin^4(k)}{(E-|\varepsilon_0(k)|+i\eta_k)^2} \frac{1}{(E+|\varepsilon_0(k)|+i\eta_k)^2}.
\end{aligned} \tag{29}
$$

The denominator can be simplified using the same techniques as above and this gives the same final result as the earlier expression for $h^{(4)}$.

## 5.1 Change in electronic structure in the single-band model

In the effective model, only the electronic target space is considered, corresponding to the lower band at zero displacement. The rest space has been integrated out. $\Sigma$ is now a scalar quantity and it is once again given by a single diagram,

$$\Sigma(k,E) = \alpha^2 V(k,E) = 4\alpha^2 t^2 \frac{\sin^2(k)}{E-|\varepsilon_0(k)|+i\eta_k}. \tag{30}$$

In this case, $\Sigma(k,E)$ is an explicit function of $E$ and it is not possible to interpret it purely as a change in the dispersion. Since the true change in the electronic structure involves a change in the orbital composition of the bands and thus coupling between the bands and changes in the wave functions, it is not possible to capture this entirely in a single-band model. However, if we restrict ourselves to the vicinity of the lower band in terms of energy, we find

$$\Sigma(k,-2t\cos k) = -t\frac{\sin^2(k)}{\cos(k)}\alpha^2, \tag{31}$$

which is equal to the exact second-order expansion of Eq. (5).

At the same time, the self-energy of Eq. (30) has a pole at $E = |\varepsilon_0|$, the energy of the upper band that has been integrated out. In the spectral function $A(E,k) = -\frac{1}{\pi}\operatorname{Im}G(E,k)$, this shows up as interaction-induced spectral-weight transfer, as shown in Fig. 5. The original spectral weight of the noninteracting, i.e., undistorted downfolded model (grey peak) is distributed to the positions of the lower and upper band of the interacting, i.e., distorted model (orange peaks). Thus, even though it cannot represent the matrix structure of the electronic Green's function, the downfolded model has spectral weight at the right locations. Note that there is no imaginary part in the self-energy and thus no additional broadening of these peaks in the downfolded model; all broadening comes from the constant $\eta = 0.05$ used for plotting the spectrum.

# 6 Constrained density-functional perturbation theory

The downfolding procedure employed above is based on an explicit resummation of the diagrammatic series and is able to reproduce the screening from bare to dressed lattice potential exactly. This approach can be applied here, since we have full knowledge of the entire electronic structure and the electron-phonon coupling. In *ab initio* calculations, the downfolding

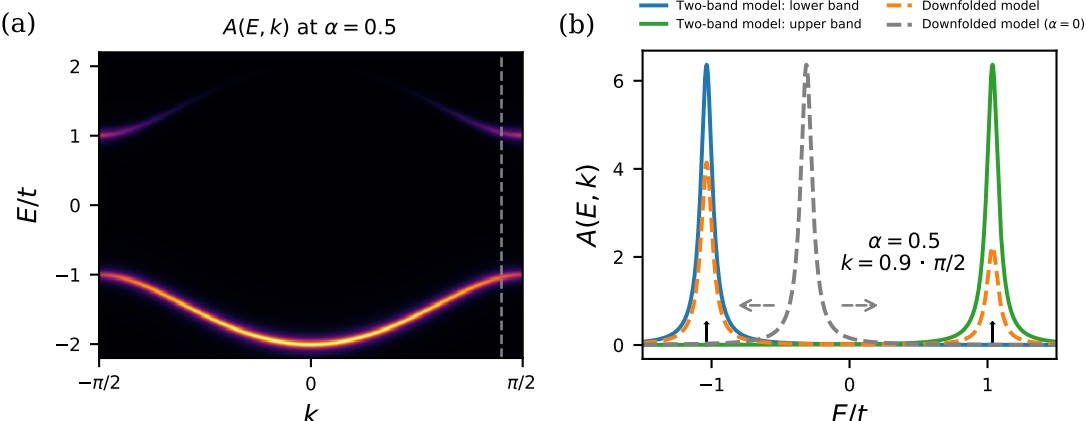

Figure 5: Spectral function of the downfolded single-band model for $\alpha = 0.5$. Here, a broadening $\eta = 0.05$ is used to improve visibility. (a) Spectral-weight transfer to the upper band occurs due to the self-energy. See Fig. 1b for the dispersion in the original two-band model. (b) Cross-section at $k = 0.9 \cdot \frac{\pi}{2}$, indicated by the grey dashed line in (a). The original model has two bands with spectral weight at $E = \pm|\varepsilon(k)|$, respectively (small vertical bars). In the single-band model at $\alpha > 0$, the self-energy leads to some spectral-weight transfer to the position of the upper band.

is usually done somewhat differently. Indeed, cDFPT is a tool commonly used for downfolding electron-phonon systems onto an electronic target space and calculating corresponding partially screened phonon frequencies. In general, it evaluates a Feynman diagram similar to Fig. 3a, with the restriction that at least one of the two electronic propagators shall not be part of the target space.

In the SSH model, if the lower band is chosen as the target space, cDFPT includes the only relevant screening process, with one $+$ and one $-$ electron, in its calculation of the partially screened phonon frequency. In other words,

$$\Pi_{m,n}^{\text{cDFPT}} = \begin{cases} 0 & \text{for } m = n = -, \\ \Pi_{m,n}(k) & \text{otherwise.} \end{cases} \tag{32}$$

Here $\Pi_{m,n}(k)$ is defined and used as in Eq. (19). In the SSH model, $\Pi_{--}$ anyway does not contribute to the phonon renormalization, and as a result the cDFPT phonon frequency is identical to the fully screened phonon frequency.

The cDFPT low-energy model then basically consists of the fully screened phonon, the lower electronic band, and *no* electron-phonon coupling, since $g_{--} = 0$. Because of this special property of the SSH model, there is no real distinction between the partially and fully screened phonon.

# 7  Breakdown of perturbation theory at half-filling

The series expansion of the potential $E(\alpha)$ around $\alpha = 0$, performed either diagrammatically or by directly taking derivatives of $\varepsilon(k, \alpha)$, shows a regular pattern. Only even powers of $\alpha$ are allowed. For a given power $\alpha^{2n}$, the diagrammatic contribution will be of the form (modulo prefactor) $g^{2n} G_-^n G_+^n$. The $2n$ electron-phonon vertices $g$ contribute $(2t)^{2n} \sin^{2n}(k)$, whereas the Green's functions can be reduced to $n(\varepsilon_-(k))/(2\varepsilon_0)^{2n-1} \propto n(\varepsilon_-(k))/\cos^{2n-1}(k)$. The only

role of the density is to determine the integration range, via $k_f$. This becomes qualitatively important for $\langle n \rangle \to 1$, $k_f \to \pi/2$, since $\varepsilon_0(k_f) \to 0$. The denominator in the integral diverges and as a result the entire integral is no longer convergent. In other words, perturbation theory around $\alpha = 0$ is not possible since $E(\alpha)$ is not an analytical function anymore. Physically, the dimerization at half-filling is a Peierls transition caused by the perfect nesting of the Fermi surface points $\pm\pi/2$ with respect to the dimerization wave vector $\pi$ (in the original Brillouin zone). Thus, at half-filling, dimerization will occur even at arbitrarily large force constant $k_s$.

## 8 Conclusion and discussion

A key question in the investigation of coupled electron-phonon systems is the evolution of the total energy and electronic structure as a function of atomic displacement. In *ab initio* studies, it is desirable to gain (perturbative) access to this energy landscape starting from the undistorted structure and a small set of relevant electronic bands. In the SSH model, it is actually possible to perform this perturbative, diagrammatic expansion analytically and to trace the performance of effective models. This both provides a unique insight into "exact downfolding" and highlights the successes and possible failures of effective models.

The bare phonons in the SSH model are entirely harmonic by definition. Thus, all anharmonic effects in the potential energy have to be created by the (linear) coupling to the electrons and the resulting electronic screening. Due to the simple structure of the model, the screening can be calculated to arbitrary order in the displacement. It reduces the energetic cost of displacements and eventually leads to a CDW transition, i.e., the appearance of a new global minimum in the energy landscape at a finite displacement. In this model, all relevant quantities can be reduced to integrals over the occupied part of the Brillouin zone.

It is also possible to downfold onto a single-band model with only half the electronic degrees of freedom of the original system. The diagrammatic structure changes due to the downfolding; the electron-phonon coupling is now dynamical and quadratic in the displacement field. Still, the analytical evaluation of the diagrams determining the energy landscape is possible and agrees with the exact result. Regarding the electronic structure, the effective single-band model only has the ability to describe spectral-weight transfer and by construction does not have the ability to describe the changes in the orbital composition of the bands as the atoms move. In the cDFPT approach, as well as in the cRPA approach to Coulomb interactions, these changes in the electronic structure are usually not considered at all.

This observation is potentially relevant for several two-dimensional transition-metal dichalcogenides. For example, monolayer 1H-TaS$_2$ has a single band crossing the Fermi level and this band consists of a combination of $d_{0,+2,-2}$ orbitals. It was already known that the electronic matrix structure is imprinted on the momentum structure of the electron-phonon coupling in *ab initio* downfolding [29] and that the resulting single-band electron-phonon model accurately describes the phonon frequencies (i.e., the energy landscape close to the undistorted structure). A similar situation, with a single composite band crossing the Fermi level, occurs in 1H-NbS$_2$ [66]. An open question is how these single-band effective models perform in the description of the true electronic structure of the distorted phase. If the distortions lead to hybridization between target and rest space, downfolded models can only capture the spectral-weight transfer. On the other hand, downfolded approaches can fully describe processes that occur entirely in the target space. Thus, fluctuation diagnostics of the electron-phonon coupling [29] can provide an answer to this question.

The SSH model in the Born-Oppenheimer approximation—as studied here—is very much a simplification of the complex reality of electron-phonon-coupling and charge-density-wave physics. We assume that the lattice is one-dimensional, that the electronic hopping amplitudes

and the bare restoring forces are linear in the displacement, that there is no electron-electron interaction, that there is a single relevant phonon mode (dimerization), and that the system is in the $T = 0$ ground state. Still, some general conclusions are possible from our work. It is possible to generate anharmonic phonon terms entirely electronically, from an initial Hamiltonian that has purely harmonic phonons. Diagrammatic expressions can be constructed for the electronic screening at and beyond the harmonic level; in the general case these will be infinite series of diagrams, but here there is only a single diagram at any order in the displacement. In the presence of multiple relevant phonons, see Appendix C, the Born-Oppenheimer energy landscape will include mode-mode coupling as well. Downfolding of the electronic space generates a new perturbation series, in which effective higher-order vertices appear naturally. Unlike in the original Hamiltonian, the vertices of the downfolded system are also dynamical (frequency-dependent). As a result, the self-energy is dynamical as well, leading to spectral-weight transfer in the downfolded model. We note that this happens even though the electrons are noninteracting. The magnitude of the self-energy in the low-energy band is approximately given by the electron-phonon coupling (between the target and the rest space) squared times the displacement squared divided by the energy separation between the low-energy and the high-energy band. This supports the natural strategy of including bands in the low-energy model that are close in energy and those that are strongly coupled to the target space via the relevant phonon modes.

**Funding information**   This work is supported by the Deutsche Forschungsgemeinschaft (DFG) through the Research Training Group Quantum Mechanical Materials Modelling (RTG 2247) and Germany's Excellence Strategy (University Allowance, EXC 2077) as well as by the Central Research Development Fund of the University of Bremen.

## A   Number of minima of $E(\alpha)$

The SSH model in the limit of large $\alpha$ is unlikely to be an accurate description of any real physics, but it is useful to establish some formal results. First of all, the triangle inequality provides us with bounds on the dispersion,

$$\max(\cos(k), \alpha |\sin(k)|) \leq \frac{|\varepsilon_\pm(k)|}{2t} \leq \cos(k) + \alpha |\sin(k)|. \tag{33}$$

Thus, in the limit of large $\alpha$, $\varepsilon(k)$ is roughly proportional to $\alpha \sin(k)$. The total energy is then dominated by the purely lattice term proportional to $k_s \alpha^2$. We conclude that the energy landscape $E(\alpha)$ is bounded from below, as it should be.

Two types of energy landscape $E(\alpha)$ are discussed in the text, one with a single minimum at $\alpha = 0$ and one with two minima at $\alpha = \pm\alpha^*$. In fact, we can proof that these are the only two possibilities, no further local minima are allowed.

First, we define the auxiliary function $f(x) = -\sqrt{1 + x^2}$, so that

$$\varepsilon_-(k, \alpha) = |\varepsilon_0(k)| f\left(\alpha \left|\frac{\sin(k)}{\cos(k)}\right|\right). \tag{34}$$

We observe that the second derivative of $f$, $f'' = -(1 + x^2)^{-3/2}$, is monotonously increasing for $x \geq 0$. This implies that $d^2\varepsilon_-(k, \alpha)/d\alpha^2$ is also monotonously increasing as a function of $\alpha$ for $\alpha \geq 0$ and the same holds for $E(\alpha)$, which is just a $k$-integral over $\varepsilon_-$. Thus, there can be at most one $\alpha \geq 0$ where $d^2E(\alpha)/d\alpha^2 = 0$. In $E(\alpha)$, local minima ($d^2E/d\alpha^2 > 0$) and local maxima ($d^2E/d\alpha^2 < 0$) alternate, so by the intermediate value theorem $d^2E/d\alpha^2$ must cross zero between every local optimum of $E(\alpha)$. This can happen only once for $\alpha \geq 0$, so there are

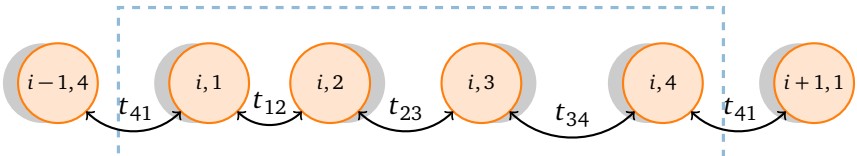

Figure 6: Length-4 unit cell with a periodic distortion (phonon eigenmode $\alpha_2$). The double arrows indicate the four hopping parameters $t_{ij}$. The atoms are labeled by their unit-cell number and their position within the unit cell.

at most two optima at $\alpha \geq 0$ and one of them is at $\alpha = 0$ by symmetry. Since $E(\alpha) \to +\infty$ for $\alpha \to +\infty$, there is either a single global minimum at $\alpha = 0$ or a local maximum at $\alpha = 0$ and two global minima at $\pm\alpha^*$.

## B  Basis transformation of the electron-phonon coupling

The electronic part is most conveniently expressed in the band basis of $\hat{\varepsilon}_0$, which is $\hat{\varepsilon}$ evaluated at $\alpha = 0$. The two eigenvalues of $\hat{\varepsilon}_0$ are $\varepsilon_{\pm,0} = \pm 2t\cos(k)$ with corresponding eigenvectors

$$\vec{v}_\pm(k) = \frac{1}{\sqrt{2}}\begin{pmatrix} 1 \\ \mp e^{-ik} \end{pmatrix}. \tag{35}$$

With the eigenvectors, we can form the transformation matrix

$$\hat{U}(k) = \frac{1}{\sqrt{2}}\begin{pmatrix} 1 & 1 \\ -e^{-ik} & e^{-ik} \end{pmatrix}, \tag{36}$$

which diagonalizes $\hat{\varepsilon}_0$. This yields the electron-phonon coupling in the band basis,

$$\hat{g}(k) = \hat{U}^{-1}(k)\underline{\hat{g}}(k)\hat{U}(k) = 2t\begin{pmatrix} 0 & i\sin(k) \\ -i\sin(k) & 0 \end{pmatrix}. \tag{37}$$

## C  Beyond dimerization: 4-site unit cell

At half-filling, the dimerization is commensurate in the sense that $2k_f = q_{\text{dimerization}}$. We have already shown that dimerization can also be energetically favorable away from half-filling, but so far we have not considered CDWs with other periodicities. In this appendix, we consider periodicity 4, which allows for the study of additional phonon modes. Because this doubling of the unit cell increases both the number of phonons and the number of electronic bands, it is more difficult to derive compact formulas and our treatment remains relatively brief, highlighting some similarities and differences to the 2-site unit cell.

In this case, it is convenient to first consider the electronic dispersion as a function of the four hopping amplitudes $t_{ij}$, as shown in Fig. 6. In the SSH model, these hopping parameters will be linear functions of the atomic displacements.

The electronic Hamiltonian is

$$\hat{\varepsilon}(k) = \begin{pmatrix} 0 & t_{12} & 0 & t_{41}\exp(4ik) \\ t_{12} & 0 & t_{23} & 0 \\ 0 & t_{23} & 0 & t_{34} \\ t_{41}\exp(-4ik) & 0 & t_{34} & 0 \end{pmatrix}. \tag{38}$$

With $t_{\mathrm{RMS}}^2 = (t_{12}^2 + t_{23}^2 + t_{34}^2 + t_{41}^2)/4$, its four eigenvalues $\varepsilon_{++}$, $\varepsilon_{+-}$, $\varepsilon_{-+}$, and $\varepsilon_{--}$ read

$$\varepsilon_{\pm\pm}(k) = \pm\sqrt{2t_{\mathrm{RMS}}^2 \pm \sqrt{4t_{\mathrm{RMS}}^4 + 2t_{12}t_{23}t_{34}t_{41}\cos(4k) - t_{12}^2 t_{34}^2 - t_{23}^2 t_{41}^2}}. \qquad (39)$$

Here, $-\pi/4 < k \le \pi/4$ is the Brillouin zone corresponding to this unit cell. As for the dimerization transition, the total electronic energy is given by $\sum_m \int dk\, \varepsilon_m(k) n(\varepsilon_m(k))$.

Now, in the SSH model, the hopping parameters depend linearly on the atomic displacements. We consider three phonon modes $\alpha_1$, $\alpha_2$, and $\alpha_3$ defined by

$$\begin{aligned}
t_{12} &= t(1 + \alpha_1 + \alpha_2), \\
t_{23} &= t(1 - \alpha_1 + \alpha_3), \\
t_{34} &= t(1 + \alpha_1 - \alpha_2), \\
t_{41} &= t(1 - \alpha_1 - \alpha_3).
\end{aligned} \qquad (40)$$

$\alpha_1$ is the dimerization mode studied in the main text, $\alpha_2$ is sketched in Fig. 6, and $\alpha_3$ is obtained from $\alpha_2$ by translating the unit cell by one atom. They are eigenmodes at $q = 0$. Combining Eqs. (39) and (40), it is possible to calculate $\varepsilon(k; \alpha_1, \alpha_2, \alpha_3)$ and its derivatives with respect to $\alpha_i$. Using computer algebra, it is possible to evaluate these derivatives straightforwardly, although the expressions quickly become unwieldy. Below, we will briefly discuss the nonzero terms at the lowest orders. Finally, integrating these derivatives of the dispersion over the filled part of the Brillouin zone (for each band) then gives the terms in the Taylor expansion of $E(\alpha_1, \alpha_2, \alpha_3)$, as in Sec. 3 of the main text. The first derivative vanishes as expected, $\partial_{\alpha_1}\varepsilon = \partial_{\alpha_2}\varepsilon = \partial_{\alpha_3}\varepsilon = 0$. The second derivative is diagonal in the phonon index, $\partial_{\alpha_1,\alpha_2}\varepsilon = \partial_{\alpha_1,\alpha_3}\varepsilon = \partial_{\alpha_2,\alpha_3}\varepsilon = 0$, so the only nonzero elements are $\partial_{\alpha_1,\alpha_1}\varepsilon$ and $\partial_{\alpha_2,\alpha_2}\varepsilon = \partial_{\alpha_3,\alpha_3}\varepsilon$. At the level of the third derivative, we find a finite term with mixed phonon labels, to be explicit:

$$\partial_{\alpha_1,\alpha_2,\alpha_2}\varepsilon = -\partial_{\alpha_1,\alpha_3,\alpha_3}\varepsilon = t\left(-\frac{\cos 2k}{\cos k}, \frac{\cos 2k}{\sin k}, -\frac{\cos 2k}{\sin k}, \frac{\cos 2k}{\cos k}\right). \qquad (41)$$

Here, the four components in the vector correspond to the bands from lowest to highest energy, and we have assumed $k > 0$. At fourth order, we find nonzero expressions only for the terms where the derivatives appear in pairs, e.g., $\partial_{\alpha_1,\alpha_1,\alpha_2,\alpha_2}\varepsilon$. Symmetries and momentum conservation still ensure that many terms in the expansion vanish, but already at the third order we see that qualitatively new terms appear compared to energy landscape for the 2-site unit cell. In other words, the Feynman diagrams studied in the main text are all relevant in general, but diagrams that were "forbidden" in that simple system can play a role. It is difficult to make any statements about the sign and relative magnitude a priori; for a computational case study of nonlinear mode-mode coupling, see Ref. [67].

Similarly, the Hamiltonian can be written in terms of the bare dispersion and the electron-phonon couplings, now as $4 \times 4$ matrices. In analogy to the main text, the terms in the expansion of $E(\alpha_1, \alpha_2, \alpha_3)$ can then be obtained diagrammatically.

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
