# Peer review of "Downfolding the Su-Schrieffer-Heeger model"

_SciPost Physics, doi:SciPost Phys. 11, 079 (2021)_

## Round 2 · Referee Report · Matthieu Verstraete (Referee 1) · 2021-6-13

Strengths

1 - the manuscript deals with an interesting and unexplored issue in the quality of downloading methods for treating electron phonon coupled problems.
2 - the analysis is very in depth and complete, given the simplicity of the SSH interactions
3 - the perturbation theory treatment is transparent, identifying orders of expansion and fully analytical expressions for self energies and effective coupling parameters
4 - novel features are brought to light, such as higher orders of phonon contribution to the effective EPI, and the distinction between spectral weight and orbital mixing effects.

Weaknesses

1 - the model's simplicity (few bands, no competing anharmonicity) are an advantage in the analysis, but it is not trivial to determine which aspects of the treatment will carry over to full ab initio downfolding calculations
2 - the full parameter space of the SSH model is not explored, only the dimerization transition. It could be that some of the derivations actually carry over to the whole space, I have little feeling for this.

Report

Comments:
The SSH model is presented as targeted for CDW transitions : there are additional ingredients in the full treatment of the CDW, in particular more complex (or purely electronic) screening effects, nesting etc... There is a (sometimes sterile) debate in the literature about the nature of CDWs (purely electronic, always with a phonon contribution, with or without nesting...).
The present authors equate CDW with Peierls and with an explicit electron phonon mechanism, but in principle CDW could arise from purely electronic instabilities which give broken translation symmetry at equilibrium ionic positions. I think it would be useful to give some context for this and recognize there are CDW cases and mechanisms which may be completely outside the SSH type of mechanism.

What happens at finite T? The electron hole symmetry might be broken, but the Green's function treatment and other derivations should be easy to transpose.

In Sect 4 the first derivative coupling is cited, and indeed is 99% of the literature, but in advanced theories (e.g. Allen Heine Cardona) second order terms can appear as well for the Fan and Debye Waller contributions to the self energy.

I do not quite understand the phrase before Eq 18 "e.g. at order alpha^2". In principle self energies can contain arbitrarily high order diagrams...

A reference is missing for Hirsch Fye QMC, though this is incidental.

Requested changes

I would appreciate some more comment on the general applicability of the different conclusions of the paper. Is the appearance of 2 phonon diagrams in the downlfolded interaction generic? It seems to depend only on +- alpha symmetry, but it could necessitate also a very simple band structure or a 1D dynamics etc...

And the reasoning about orbital mixing out of the target space is insinuated to be general, but a stronger statement might be made. Do the authors suggest a criterion for choosing the best target space in a full ab initio calculation? (adding a few bands which might mix in, for instance)

The dimerization transition is enforced explicitly. Could the authors
1 - provide an overview of other known phases of the SSH model?
2 - make a statement about which aspects of the calculations will carry over to other phonon modes, band structures etc...?
3 - comment on the relation between the induced anharmonicity and one that would be present in the original system? Will they just sum or somehow interfere?

typos etc:
bottom of p.2: "despite the unquestionable"
fig 2: the x axis labels overlap with the caption

---

## Round 3 · Referee Report · Matthieu Verstraete (Referee 1) · 2021-8-25

Report

I appreciate the authors' revisions, and understand a bit more clearly the 2-3 points which they will not address in the present paper (temperature, more detailed relation to realistic models, etc...)

I would suggest to add an estimated temperature and energy scale (it is probably obvious from phonon energies etc...) below which the downfolded model is expected to work
  • validity: top
  • significance: high
  • originality: high
  • clarity: high
  • formatting: excellent
  • grammar: perfect

Author:  Erik van Loon  on 2021-09-03  [id 1733]

(in reply to Report 1 by Matthieu Verstraete on 2021-08-25)
Category:
answer to question

The $T=0$ dowfolding formalism developed in this manuscript remains applicable at small temperatures if the temperature is small compared to the relevant electronic and lattice energy scales. For the electronic degrees of freedom, the assumption is that the band minimum of the high-energy band is sufficiently far from the Fermi level, $k_B T << E(\Gamma)-\mu$.

For the lattice degrees of freedom, it is important to remember that we are dealing with atoms with ‘infinite mass’, i.e., without kinetic energy. In that case, and with the displacement as a classical variable, at finite temperature we need to consider an ensemble of configurations with different displacements, with their relative probably given by the appropriate Boltzmann factor. For a quadratic potential $½ \omega^2 \alpha^2$, where $\alpha$ is the displacement, equipartition states that $k_b T/2 = ½ \omega^2 <\alpha^2>$, so the typical displacement has magnitude $|\alpha|=\sqrt(k_B T/\omega^2)$. There is a definite band structure/spectral function corresponding to any displacement $\alpha$, their difference is given by $g_k |\alpha|=g_k \sqrt{(k_B T/\omega^2)}$, where $g_k$ is the electron-phonon coupling. Thus, the thermal averaging leads to an averaging over slightly displaced spectal functions due to electron-lattice coupling. In other words, the thermal averaging over lattice degrees to freedom leads to broadening of the spectrum even when we are dealing with infinite mass atoms. On the other hand, if we would incorporate the finite mass of the nuclei, phonon-induced spectral broadening takes places even at $T=0$ and can be approximately calculated using the usual Feynman diagrams.

Finally, in the manuscript we identify the $T=0$ phase transition purely based on the minimum of the energy. At finite temperature, the energy gain of the dimerized phase needs to be large compared to the temperature to see a phase transition.

---

## Round 3 · Referee Report · Anonymous (Referee 2) · 2021-9-9

Report

In the present manuscript the authors investigate the downfolding of the Su-Schrieffer-Heeger model as a minimal model for electron-phonon coupling in order to identify relevant effects and mechanisms.
The peculiarity of this model lies in the fact that a periodic displacement of the atoms can open a band gap and thereby lower the total energy of the system leading to a CDW transition. It is particularly suitable for the investigation of downfolding since it is possible to perform all calculations exactly once the Born-Oppenheimer approximation has been applied.
The comparison of the direct calculation of properties of the SSH model in the Born-Oppenheimer approximation at finite displacement with a perturbative diagrammatic expansion around the undistorted state à la DFPT is shown to correctly capture the renormalizations of the phonon frequency and the electronic structure by the electron-phonon coupling. At the same time, the effective single-band model for the dimerization transition in the SSH model with a dynamical interaction between an electron and two phonons is shown to faithfully reproduce the energy landscape and the CDW.

The importance of the present work lies in addressing a key question in the investigation of coupled electron-phonon systems on the evolution of the total energy and electronic structure with the atomic displacement. In the SSH model it is possible to perform the diagrammatic expansion analytically and to trace the performance of effective models, providing an important insight into the reliability of downfolding as well as the use of effective models.
Due to the simple structure of the model, the screening can be calculated to arbitrary order in the displacement. In particular, it is possible to take into account changes in the electronic structure which are usually not considered in the standard cDFPT or cRPA approach. Despite the simplicity of the model, one learns some general features of the downfolding and for the specific phononic problem, that it is possible to generate anharmonic phonon terms entirely electronically, from an initial Hamiltonian that has purely harmonic phonons.

The paper on this timely question is clearly written and I recommend publication in SciPost Physics.

---

## Round 3 · Author Response

Dear editor,

We would like to thank the referee for his report and we are pleased to see that he is overall positive about our manuscript. We agree with the referee's assessment of the strengths and weaknesses of our manuscript: Our overarching purpose was to study an exactly solvable example of electron-phonon interactions and downfolding. Our choice of the dimerization transition in the SSH model in the Born-Oppenheimer approximation makes that possible. It should be seen as a minimal model to elucidate some interesting and important aspects of the physics, not as a fully realistic treatment of what happens in specific materials. Indeed, the current model-based work is largely complementary to the research line of ab initio studies of electron-phonon coupling, where the full complexity is addressed.

From the referee report, it is clear that we should discuss in more detail how the results derived in this simple model translate to more generic electron-phonon systems. In the revised manuscript, we have addressed this point in several places. Our three main approximations are the classical description of the phonons (Born-Oppenheimer), the restriction to a single phonon mode (dimerization), and considering a Hamiltonian with only the lowest-order terms in the atomic displacements (the SSH model). In the revised manuscript, we explain in more detail how these three approximations lead us to our results. We have also included a brief discussion of additional phonon modes in a length-4 unit cell, still within the Born-Oppenheimer approximation of the SSH model. Since the matrix size of all objects quickly increases with the number of atoms in the unit cell, making the expressions unwieldy, we restrict ourselves to the calculation of the electronic energy in this enlarged cell, and do not perform the full downfolding procedure. This topic could be interesting for future work.

In the literature, two different conventions exist for coordinates in the SSH model Hamiltonian: the hopping term can be written either as $1+ũ_i $ or as $1+u_{i}-u_{i+1}$. The former version was used in the original submission of our manuscript and is frequently used in the discussion of topology in the SSH model. On the other hand, the second convention is more intuitive in terms of atomic coordinates, since only the distance between atoms has physical meaning. The two conventions are related by the linear transformation $ũ_i = u_{i} - u_{i+1}$, which also changes the prefactor of the bare phonon term. To deal with displacements beyond dimerization, i.e., the length-4 unit cell, the second version is much more natural. Because of this, the revised manuscript uses the second convention throughout, which leads to different prefactors in some of the equations (see list of changes below), but does not change the physics.

Below, we discuss the points made in the report and the requested changes one by one. With these changes, listed at the end of this document, we believe our manuscript is suitable for publication in SciPost Physics.

Yours sincerely,

The authors

Expand our discussion on CDWs beyond the Peierls type

The SSH model is presented as targeted for CDW transitions : there are additional ingredients in the full treatment of the CDW, in particular more complex (or purely electronic) screening effects, nesting etc... There is a (sometimes sterile) debate in the literature about the nature of CDWs (purely electronic, always with a phonon contribution, with or without nesting...). The present authors equate CDW with Peierls and with an explicit electron phonon mechanism, but in principle CDW could arise from purely electronic instabilities which give broken translation symmetry at equilibrium ionic positions. I think it would be useful to give some context for this and recognize there are CDW cases and mechanisms which may be completely outside the SSH type of mechanism.

This is absolutely correct: The SSH model has a CDW transition, but that does not mean that all CDWs are of this nature. We have added a paragraph to the introduction explaining that non-SSH CDWs also occur. Our interest here is in the SSH model, since we can solve it, and we try to identify effects that are of general use.

Finite T

What happens at finite T? The electron hole symmetry might be broken, but the Green's function treatment and other derivations should be easy to transpose.

There are two aspects of temperature that can be discussed:

First, the electronic temperature makes the Fermi-Dirac distribution function smooth. This can be incorporated relatively straightforwardly in diagrammatic theories, for example by using Matsubara frequencies. This makes the momentum integrals somewhat more complicated, but numerical treatment should still be possible. One qualitative change is that it now starts to matter if one performs calculations at constant electron density or at constant chemical potential ($k_f$), since the total density is now an explicit function of μ, T and α.

This paper is aimed towards the investigation of electronic downfolding based on density-functional-theory calculations, and T≈0 is usually assumed from the start in downfolding procedures. Because of this, we believe that the investigation of electronic temperature effects does not fit so well into the scope of this paper. We do believe that it might be an interesting topic for future work.

A second aspect is the thermal occupation of phonons. Within the Born-Oppenheimer Approximation, the thermal excitation of lattice vibrations is captured if the potential energy surface is described correctly, as is the case in our downfolding approach. Effects that go beyond the Born-Oppenheimer approximation and mix target with rest space states are beyond the applicability range of the downfolding approach.

2nd order electron-phonon coupling

In Sect 4 the first derivative coupling is cited, and indeed is 99\% of the literature, but in advanced theories (e.g. Allen Heine Cardona) second order terms can appear as well for the Fan and Debye Waller contributions to the self energy.

Here, we are considering the lattice degrees of freedom in the Born-Oppenheimer approximation. In other words, we fix all displacements and then diagonalize the resulting tight-binding Hamiltonian. This leads to a change in electronic eigenstates and eigenenergies due to displacements, but not to usual self-energy effects such as a finite lifetime/broadening. Diagrammatically, the Born-Oppenheimer approximation means that there is no phonon propagator (the classical phonons only appear as external lines) and no second-order Fan or Debye-Waller diagrams appear. The diagrammatic structure is equivalent to that of non-interacting electrons in a Zeeman field or subject to lattice inhomogeneities.

Minor things

I do not quite understand the phrase before Eq 18 "e.g. at order $\alpha^2$". In principle self energies can contain arbitrarily high order diagrams...

By phonon self-energy, we mean diagrams with two external lines, as usual. In this case, as discussed above, there is no phonon propagator due to the Born-Oppenheimer approximation. This means that α is only allowed to appear as an external line, leading to the statement in the text. We have rewritten the statement in the text to make this point clearer.

A reference is missing for Hirsch Fye QMC, though this is incidental.

A reference to the original article has been added.

Requested changes

Appearance of 2 phonon diagrams in the downfolded interaction

Is the appearance of 2 phonon diagrams in the downfolded interaction generic? It seems to depend only on +- α symmetry, but it could necessitate also a very simple band structure or a 1D dynamics etc...

The appearance of a 2-phonon interaction is generic, in the sense that one should expect it to appear when downfolding from arbitrary Hamiltonians. In fact, one should generally expect to generate all possible interactions in the downfolded model. The generation of interactions of arbitrary orders is usual for procedures where degrees of freedom are integrated out. Diagrammatically, the downfolded interactions are obtained by drawing all possible diagrams with a given number of external lines, so specific interactions can only vanish if symmetries forbid certain classes of diagrams or if all diagrams with a given number of external lines would sum to zero. The first situation occurs here and forbids all interactions except the two-phonon interaction. We have expanded the discussion on this at the start of Section 5.

The appearance of vertices with arbitrary order also occurs in, e.g., the functional renormalization group, Kadanoff decimation in the Ising model, or the dual fermion expansion around dynamical mean-field theory.

Criterion for the best target space in ab initio calculations

And the reasoning about orbital mixing out of the target space is insinuated to be general, but a stronger statement might be made. Do the authors suggest a criterion for choosing the best target space in a full ab initio calculation? (adding a few bands which might mix in, for instance)

In this model, we find that the self-energy in the single-band model due to the effective interaction induced by high-energy states that are integrated out is proportional to the electron-phonon coupling between these bands squared, divided by the energy separation/barrier. In the present case, that expression is exact, in general one also expects this formula from second-order perturbation theory (e.g., $J \sim t^2/U$ in the Hubbard model, or the expression $t^2 G$ for electronic hybridization), with possible corrections at higher orders. So the target space should consist of states that are close in energy and/or strongly coupled, whereas states further away in energy or weakly coupled can be integrated out relatively safely.

One aspect of this that we discuss in more detail in the revised manuscript is retardation. In electron-phonon problems, one is used to retarded interactions coming from the phonons. Here, however, the Born-Oppenheimer approximation rules out this kind of retarded interaction. Instead, the energy-dependence (retardation) occurs due to the integrating out of high-energy electronic states.

Calculations beyond dimerization

The dimerization transition is enforced explicitly. Could the authors

  • provide an overview of other known phases of the SSH model?
  • make a statement about which aspects of the calculations will carry over to other phonon modes, band structures etc...?
  • comment on the relation between the induced anharmonicity and one that would be present in the original system? Will they just sum or somehow interfere?

Dimerization is the favored phase at half-filling, since it is commensurate with $2k_f$ in that case. Away from half-filling, the expectation at $T=0$ is that a phase transition to an ordered phase is likely to set in. However, determining the ground state at arbitrary filling and allowing for all forms of symmetry breaking is not so easy, especially when dealing with incommensurate order or with charge density waves with large unit cells.

In the present formalism, it is conceptually relatively straightforward to enlarge the unit cell to study other CDWs, but the calculations lose a great deal of transparency. As an example, in the appendix of the revised manuscript we consider the doubling of the unit cell to length 4. This doubles the number of bands and allows for the study of 3 different displacement modes instead of the single mode considered in the original manuscript. As a result, the Born-Oppenheimer total energy becomes a function of three variables, $\alpha_i$. The series expansion of $E(\alpha_i)$ along the direction of the dimerization mode does not change (i.e., the terms with $\alpha_{dimerization}$), but off-diagonal terms also appear. It is a priori difficult to state anything about the sign (and magnitude) of these terms. Indeed, Figure 2(a) already shows that $\Delta\omega^2$ and $h^(4)$ have opposite sign in the SSH model, and in Appendix A we have shown that the electronic screening term is bounded by $\sqrt{ \alpha^2}$ at large $\alpha$, which is only possible if the terms in the Taylor expansion do not have a fixed sign.

Regarding the relation between intrinsic and induced anharmonicity: As the reviewer correctly observed, the original system, i.e., the bare phonons, are harmonic in this model. The anharmonicity is induced by the displacements, which lead to electronic screening of the phonons and thus the anharmonic terms appear in the Taylor expansion, see Eqs. 8-13 of the manuscript. However, in general the bare phonons could be anharmonic. The phononic Hamiltonian could include third- or higher-order terms, which are called three-phonon or N-phonon processes. These terms will contribute to the induced anharmonicity as a sum in the form of Feynman diagrams with arbitrary sign.

---

## Round 3 · List of Changes

Change in Introduction:

Despite ~~of~~ the unquestionable success of the current ab initio computational methods, ...

Change in Introduction:

Additionally it was applied to monolayer 1H-TaS_2 ...

Change in Introduction:

Previous investigations using this model have studied properties such as the effective mass [57,58] and the band structure [59,60], but also phonon-related properties [61] ~~in this model~~.

Change in Introduction:

In some sense, this is similar to the method employed in Hirsch-Fye Quantum Monte Carlo [63], where a Hubbard-Stratonovich transformation is used ...

New paragraph in Introduction:

We choose to study the SSH model here for its simplicity, acting as a minimal model for electron-phonon coupling. At the same time, this means that there are many relevant aspects of electron-phonon coupling and CDWs that are not captured by the SSH model. In particular, the SSH model neglects Coulomb interactions between the electrons, and these are responsible for important effects such as screening and entirely electronic CDWs without lattice displacement. Furthermore, in higher dimensions, the shape of the Fermi surface can play an important role, in the form of nesting and Van Hove singularities. Given the complexity of electron-phonon systems, studying simple models is a useful way to identify relevant effects and mechanisms.

Change of notation in Sec. 2:

We have changed the notation in Sec. 2, changing the Hamiltonian in Eq. 1: Old: H = -t \sum_{i=0}^{N-1} (1+u_i) (c^\dagger_i c_{i+1}+c^\dagger_{i+1} c_{i}) + \frac{k_s}{2} \sum_{i=0}^{N-1} (u_{i+1}-u_{i})^2. ```` New: H = -t \sum_{i=0}^{N-1} (1+u_i-u_{i+1}}) (c^\dagger_i c_{i+1}+c^\dagger_{i+1} c_{i}) + \frac{k_s}{2} \sum_{i=0}^{N-1} (u_{i+1}-u_{i})^2. ```` This leads to several changes in factors 2 and 1/2 in the equations in the text of Sec. 2 and 4 and in Eqs. 2, 3, 4, 7 and 11. We have also rescaled the parameters in Figure 2 and changed the numbers in the caption to reflect this.

New sentence in Sec. 3:

In Appendix A, we show that there can be at most two minima, symmetrically located around &\alpha;=0. Only even orders of &\alpha; appear due to the symmetry of the system.

Change in Sec. 4.1:

For the phonon self-energy, i.e., ~~at order α^2~~ with two external lines, there is only a single diagram, ...

New sentence in Sec. 5:

... and an infinite set of vertices can appear in this way. The only general constraints are the conservation of the fermion number and momentum conservation. Thus, the low-energy Hamiltonian can contain interactions of the form α^m (c^\dagger c)^n for arbitrary m and n. However, additional symmetries of the system can provide further constraints on the effective action.

New paragraph in Sec. 5:

The downfolded SSH model has only a single effective interaction vertex. This happens because the electron-phonon coupling in the original SSH model only has a single external high-energy electron (blue line in Fig.~4a. On the other hand, if the original model had contained either electron-electron interactions in the high-energy band or electron-phonon coupling between different electronic states in the high-energy band, then the downfolding would be more involved, since more diagrammatic contributions would appear in the expression for the effective action.

Changes in Discussion:

For example, monolayer 1H-TaS$_2$ has a single band crossing the Fermi level ... A similar situation, with a single composite band crossing the Fermi level, occurs in ~~monolayer~~ 1H-NbS_2 [66].

New paragraph in Discussion:

The SSH model in the Born-Oppenheimer approximation - as studied here - is very much a simplification of the complex reality of electron-phonon-coupling and charge-density-wave physics. We assume that the lattice is one-dimensional, that the electronic hopping amplitudes and the bare restoring forces are linear in the displacement, that there is no electron-electron interaction, that there is a single relevant phonon mode (dimerization), and that the system is in the T=0 ground state. Still, some general conclusions are possible from our work. It is possible to generate anharmonic phonon terms entirely electronically, from an initial Hamiltonian that has purely harmonic phonons. Diagrammatic expressions can be constructed for the electronic screening at and beyond the harmonic level; in the general case these will be infinite series of diagrams, but here there is only a single diagram at any order in the displacement. In the presence of multiple relevant phonons, see Appendix C, the Born-Oppenheimer energy landscape will include mode-mode coupling as well. Downfolding of the electronic space generates a new perturbation series, in which effective higher-order vertices appear naturally. Unlike in the original Hamiltonian, the vertices of the downfolded system are also dynamical (frequency-dependent). As a result, the self-energy is dynamical as well, leading to spectral-weight transfer in the downfolded model. We note that this happens even though the electrons are noninteracting. The magnitude of the self-energy in the low-energy band is approximately given by the electron-phonon coupling (between the target and the rest space) squared times the displacement squared divided by the energy separation between the low-energy and the high-energy band. This supports the natural strategy of including bands in the low-energy model that are close in energy and those that are strongly coupled to the target space via the relevant phonon modes.

Changes in Appendix B:

Clarified equations: Function arguments (k) are written explicitly, additional \hat on U.

New Appendix C on 4-site unit cell.

The situation in a 4-site unit cell is discussed briefly in this appendix.

---

## Editorial Decision

published